# Shelf-Life and Quality of Anchovies (*Engraulis encrasicolus*) Refrigerated Using Different Packaging Materials

Angela Dambrosio [1], Nicoletta Cristiana Quaglia [1], Maria Antonietta Colonna [2], Flavia Capuozzo [1], Francesco Giannico [1], Simona Tarricone [2,*], Anna Caputi Jambrenghi [2] and Marco Ragni [2]

1   Department of Veterinary Medicine, University of Bari Aldo Moro, 70010 Bari, Italy; angela.dambrosio@uniba.it (A.D.); nicolettacristiana.quaglia@uniba.it (N.C.Q.); flavia.capuozzo@uniba.it (F.C.); francesco.giannico@uniba.it (F.G.)
2   Department of Soil, Plant and Food Science, University of Bari Aldo Moro, 70125 Bari, Italy; mariaantonietta.colonna@uniba.it (M.A.C.); anna.caputijambrenghi@uniba.it (A.C.J.); marco.ragni@uniba.it (M.R.)
*   Correspondence: simona.tarricone@uniba.it

**Abstract:** The identification of sustainable materials as an alternative to plastic is fundamental for the protection of the environment and for the safeguard of marine ecosystems. The replacement of plastic with organic materials for fish storage might affect its shelf-life due to the high concentration of oxidizable fatty acids in fish flesh. This study investigated the effectiveness of two organic packaging materials, namely coconut fiber (Coc) and paper (Pap), as alternatives to the conventionally used polystyrene (Pol), on the shelf-life, hygienic parameters, and quality traits of anchovies refrigerated for one (T0), three (T1), or seven (T2) days. The hygienic quality of anchovies packaged in the organic Pap and Coc boxes was better than the traditional Pol, probably due to the higher porosity of the packaging, which allows a higher bacterial proliferation. Results of the T-BARS test showed a lower ($p < 0.05$) malondialdehyde concentration in fillets from the Pap group, which increased ($p < 0.05$) from the first to the seventh day of storage. The polyunsaturated fatty acids concentration did not differ between groups, but it decreased ($p < 0.05$) from T0 to T2. Coconut fiber and paper are worthy of further investigation since these packaging materials did not affect the shelf-life and fatty acid profile of anchovies during storage.

**Keywords:** anchovies; organic packaging; coconut fiber; paper; polystyrene; hygienic parameters; fatty acid profile; fillet quality; T-BARS

**Key Contribution:** Coconut fiber and paper may be successfully used for fish handling and storage since these packaging materials did not affect the shelf-life and fatty acid profile of anchovies during storage. The use of alternative materials to plastic may exert many benefits capable of contributing to the reduction of pollution in marine ecosystems.



## 1. Introduction

The removal of marine plastic litter on a large scale is expensive and unsuccessful. Therefore, a synergic collaboration between governments, research institutions, industries, and the fishery and aquaculture sector is strongly needed in order to redesign products, aiming to reduce plastic waste and to find alternative materials, such as biodegradable organic ones, possibly derived from agro-industrial by-products (fiber, starch, cellulose, resins, fungi) or from livestock production (keratin, fibroin, casein, wool, silk). Nowadays, plastic containers are widely used in the operations of fish handling and storage in the supply chain of fishing and aquaculture.

In the last five decades, the plastic industry has been growing in all the world, reaching almost 370 million tons of plastic in 2019, among which 58 million tons were produced in Europe [1]. The huge production of this synthetic organic polymer made from petroleum

is due to its numerous properties, which are useful in a wide variety of applications in everyday life, such as packaging, building, household, furniture, textiles, sports equipment, vehicles, electronics, and agriculture [2]. On the other hand, being poorly biodegradable, plastic is a very powerful source of pollution that threatens food safety and quality, human health, coastal tourism, and contributes to climate change. If discarded improperly, plastic waste can harm the environment and biodiversity. It has been estimated that about 14 million tons of plastic end up in the seas every year [3,4]. Plastic debris and waste represent a major concern for marine ecosystems and shorelines in every continent, with a higher concentration in proximity of crowded tourist destinations and high-density populated areas.

Under the influence of solar UV radiations, wind, currents, and other natural factors, plastic breaks down into small particles called microplastics (MPs, <5 mm in diameter) or nanoplastics (NPs, <100 nm in diameter) [5]. Besides the visible impacts of plastic debris on the ingestion, swimming ability, suffocation, entanglement, lacerations, and infection of hundreds of marine species, MPs are particularly harmful since they are easily ingested. Several authors reported the presence of MPs in the digestive system of various fish species, detected by means of direct visual sorting [6–8]. Dehaut et al. [9] found the presence of MPs in the 45% of biota caught in the Adriatic Sea while Pellini et al. [10] observed MPs in the 95% of the benthic flatfish *Solea solea* from the same sea. Moreover, several chemicals used in the production of plastic materials are known to exert carcinogenic effects and to interfere with the body's endocrine system, causing developmental, reproductive, neurological, and immune disorders in both humans and wildlife [11,12].

The chemical and fatty acid composition of fish fillets affect the organoleptic features of fish along with its healthiness [13]. Fillets of the pelagic species are characterized by a high nutritional value since they are rich in n − 3 fatty acids, particularly eicosapentaenoic (EPA) and docosahexaenoic (DHA) fatty acids, amino acids, and micronutrients that exert a protective effect on human health [14]. Due to the high amount of essential fatty acids, the consumption of this species is strongly recommended, especially for preventing and treating cardiovascular diseases, dementia, mental decline, and many age-related diseases in humans [15–17].

*Engraulis encrasicolus* (*European anchovy*, Linnaeus 1758) is the most important species among the pelagic fishes in the Mediterranean Sea, accounting for approximately 50% of the total [18]. Anchovies are usually classified as oily fish, due to the high content of fatty acids; the lipid fraction contains approximatively 30–40% of aliphatic long-chain fatty acids (C13-C23) that are polyunsaturated, such as EPA (C20:5n3) and DHA (C22:6n3) [19].

Oxidized fatty acids are the principal cause of food rancidity and their reaction with other metabolites determines the deterioration of the nutritional value of foods. The rancidification process is due to the hydrolysis of fats into short-chain aldehydes, ketones, and free fatty acids that are responsible for taste and odor alteration [20]. Initial oxidation produces aldehydes and ketones, and malondialdehyde (MDA) is the aldehyde related to the degree of lipid oxidation measured by the thiobarbituric acid test [21]. In particular, the double bonds of unsaturated fatty acids can be cleaved by free radical. Although fish is held under refrigeration, the polyunsaturated fat will continue to oxidize and become rancid [19].

In Italian fisheries, the most common practice for small pelagic species is to chill fish on board in polystyrene containers, which are resistant and easy to handle. Polystyrene has an estimated average life of approximately 1000 years, it is non-biodegradable, and it contributes to plastic pollution if released into the sea [2].

Any replacement of plastic with an organic packaging material for fish storage must be thoroughly studied since it might influence the storage milieu, thus affecting fish potential shelf-life, due to oxidation processes [22].

The aim of this study was to investigate the effectiveness of two organic packaging materials for fish, coconut fiber (Coc) and paper (Pap), as alternatives to the conventionally

used polystyrene (Pol), on the shelf-life, hygienic parameters and quality traits of anchovies during refrigerated storage for one (T0), three (T1), or seven (T2) days.

## 2. Materials and Methods

### 2.1. Sample Preparation

Anchovies (*Engraulis encrasicholus*) were caught in May 2021 in the central Adriatic Sea (43°21′34.0″ N; 14°34′22.7″ E) and were kept on board in 3 types of boxes with ice, each containing approximately 10 kg of product. Three lots, with 3 replicates each, were studied, consisting of anchovies stored in two organic packaging materials containing Paper (Pap; UnaxTutte, CCM Packaging, Modena (MO), Italy; 48 × 33 × 8 cm, 180 g) or Coconut fiber (Coc; Infinity mod 6VA032P, Jpack, Val Brembilla (BG), Italy; 50 × 30 × 8 cm, 220 g) versus traditional Polystyrene boxes (Pol; TKM, Gioia del Colle (BA), Italy; 49 × 33 × 11, 180 g). The time elapse from fish catching to processing on laboratory was approximately 12 h. On arrival, all lots were re-iced and kept until they were analyzed at 0 °C (±0.5 °C). The re-icing was performed periodically as required in each lot.

Anchovies were submitted to biometric measurements. Values reported as means and standard deviations of weight, length, and width [23] were 8.32 ± 0.98 g, 94.71 ± 3.40 mm, and 14.81 ± 0.82 mm, respectively. Analyses were conducted after 1 (T0), 3 (T1), and 7 (T2) days of storage on different fish batches. These intervals were chosen based on several studies evidencing that the endpoint of anchovy edibility, stored at 0 °C under ice, falls between 6 and 8 days [24,25].

### 2.2. Microbiological Analysis

For the microbiological analysis of samples stored at 0 °C on ice, the counts of total aerobic mesophilic bacteria at 30 °C (CAMT), total aerobic psychrophilic bacteria (CAPT) at 4°C, enterobacteria (EC), *Escherichia coli*, and *Pseudomonas* spp. at 1 (T0), 3 (T1), and 7 (T2) days of storage were evaluated.

For CAMT and CAPT, 30 g of the aseptically collected sample was added to 270 mL of Buffered Peptone Water (BPW) (Liofilchem, Teramo, Italy), homogenized in a stomacher (Lab-Blender 400, PBI, Milan, Italy), decimally diluted, and pour plated on Plate Count Agar (PCA) (Liofilchem, Teramo, Italy). The bacterial culture was incubated at 32 ± 1 °C for 72 ± 3 h and at 6.5 ± 1 °C for 10 days.

For *Enterobacteriaceae* count, 30 g of aseptically collected sample was added to 270 mL of BPW and homogenized in a stomacher, decimally diluted, and pour plated on Violet Red Bile Glucose Agar (VRBG), (Liofilchem, Teramo, Italy) and incubated at 37 ± 1 °C for 24 ± 3 h for 10 days.

As for the count of *E. coli*, 30 g of sample was added to 270 mL BPW and log dilutions were prepared. 1 mL of each dilution was poured into Tryptone Bile X-glucuronide Agar (TBX) and incubated at 44 ± 1 °C for 24 h. After the incubation period, the typical blue-green colonies were counted.

*Pseudomonas* spp. count was performed using 10 g of the sample added to 90 mL of Peptone Water. Log dilutions were prepared, and 0.1 mL of each dilution was applied to CFC-Pseudomonas Agar Base (Liofilchem, Teramo, Italy) and incubated for 44 + 4 h at 25 ± 1 °C. After incubation, 5 colonies with typical characteristics were taken from each plate of the two dilutions and subjected to the oxidase test. Colonies that tested positive for oxidase were confirmed as *Pseudomonas* spp.

### 2.3. Fatty Acid Profile of Anchovies Fillets

On the days of analysis, fillets were skinned, chopped, combined in a pool for each group, and homogenized. Total lipids were extracted from the homogenized samples of fish fillets (10 g) using the chloroform/methanol method described by Folch et al. [26]. Fatty acids were methylated using KOH and methanol (solution 2N) [27], and the fatty acid profile was assessed using a gas chromatograph (Shimadzu GC-17A, Kyoto, Japan) with a silicate glass capillary column (70% Cyanopropyl Polysilphenylenesiloxane BPX 70 of SGE

Analytical Science, length = 50 m, internal diameter = 0.22 mm and film thickness = 0.25 m). The temperature program was as follows: 135 °C for 7 min, followed by increases of 4 °C per minute up to 210 °C and held for 2 min. The injector and detector temperatures were set at 245 and 280 °C, respectively. Helium gas was used as a carrier with a flow of 1.2 mL/min. Fatty acids were identified using a mixture of standard fatty acids (Restek Corporation, Bellefonte, PA, USA) and were expressed as percentage (wt/wt) of total methylated fatty acids.

The food risk factors of fish were determined by calculating the atherogenic (AI) and thrombogenic (TI) indices [28]:

$$AI = [(C12\!:\!0 + 4 \times C14\!:\!0 + C16\!:\!0)] \div [\sum MUFA + \sum n-6 + \sum n-3]$$

$$TI = [(C14\!:\!0 + C16\!:\!0 + C18\!:\!0)] \div [0.5 \times \sum MUFA + 0.5 \times \sum n-6 + 3 \times \sum n-3 + \sum n-3/\sum n-6]$$

where $\sum MUFA$ is the total monounsaturated fatty acids.

### 2.4. Lipid Oxidation of Anchovies Fillets

Lipid oxidation was evaluated in anchovy samples at the arrival to the laboratory (T0) and after storage for 3 (T1) and 7 (T2) days by measuring the concentration of 2-thiobarbituric acid reactive substances (T-BARS) and expressed as mg malondialdehyde (MDA)/kg fillet [29]. The test is based on a spectrophotometric quantification (532 nm) of the reaction occurring between MDA and thiobarbituric acid (TBA), in conditions of high temperature and low pH, which determine the formation of a pink colored complex.

### 2.5. Statistical Analysis

A mixed model for repeated measures [30] was used to detect a possible relation between the packaging material (M: Pol; Pap and C) and the storage period of anchovies (S: T0 = 1 d; T1 = 3 d and T2 = 7 d) on TBARS values and fatty acid profiles of anchovy fillets. Microbiological data were converted to logarithms of the number of colonies forming units (log cfu g_1). The average and standard deviations of microbial counts were determined from the average of three replicates. Two-way analysis of variance (ANOVA) was performed to evaluate the difference in microbial counts in the different treatments and at different times using the SAS software [30]. The values were considered significant at $p \leq 0.05$ and presented as least squares means and pooled SEM values. Means were compared by Duncan's multiple range test.

### 3. Results and Discussion

### 3.1. Microbiological Analysis

The results concerning CAMT, CPAT, total *Enterobacteria*, *E. coli*, and *Pseudomonas* spp. in the Pol, Pap, and Coc samples are described in Table 1. The CAMT value remained constant during the seven-day storage period and no significant differences between packaging groups were found.

For the Pap group, mean CAPT values significantly ($p < 0.05$) increased during the storage period, from an average value of $1.53 \pm 0.14$ log cfu $g^{-1}$ on the first day to $3.11 \pm 0.09$ log cfu $g^{-1}$ at the end of the storage period. Similarly, group Coc showed mean CAPT values up to T2 ($1.65$ log cfu/$g^{-1}$) with an increase of $1.4$ log cfu/$g^{-1}$ at seven days (Table 1).

At day 7, the Pol group showed a significantly ($p < 0.05$) higher CAPT value ($4.34$ log cfu $g^{-1}$) as compared to both Pap and Coc groups.

**Table 1.** Average values (log cfu g$^{-1}$) of total mesophilic aerobic bacteria, total aerobic psychrophilic bacteria, total Enterobacteria, *E. coli* and *Pseudomonas* spp. in anchovies in relation to different packaging materials and storage times.

| Packaging Material * | Storage Time (Days) | | | | | | | | | | | | | | |
|---|---|---|---|---|---|---|---|---|---|---|---|---|---|---|---|
| | T0 (1 d) | | | | | T1 (3 d) | | | | | T2 (7 d) | | | | |
| | CAMT [1] | CAPT [2] | ENTERO [3] | E.c. [4] | PSEUDO [5] | CAMT [1] | CAPT [2] | ENTERO [3] | E.c. [4] | PSEUDO [5] | CAMT [1] | CAPT [2] | ENTERO [3] | E.c. [4] | PSEUDO [5] |
| Pap | 2.40 ± 0.11 | 1.53 ± 0.14 | 0 | 0 | 1.08 ± 0.02 | 3.30 ± 0.35 | 1.18 ± 0.25 | 1.70 ± 0,16 | 0 | 2.60 ± 0.39 | 3.18 ± 0,20 | 3.11 ± 0,09 | 0 | 0 | 3.15 ± 0.21 |
| Coc | 2.52 ± 0.11 | 1.65 ± 0.11 | 0 | 0 | 1.08 ± 0.02 | 3.04 ± 0.27 | 1.65 ± 0.07 | 1.85 ± 0,1 | 0 | 2.70 ± 0.30 | 3.32 ± 0,45 | 3.06 ± 0.10 | 0 | 0 | 3.26 ± 0.28 |
| Pol | 3.18 ± 0.61 | 1.88 ± 0.04 | 0 | 0 | 2.36 ± 0.15 | 3.76 ± 0.08 | 2.74 ± 0.09 | 2.9 ± 0,15 | 0 | 2.65 ± 0.35 | 3.79 ± 0,14 | 4.34 ± 0.32 | 2.99 ± 0.18 | 0 | 3.51 ± 0.04 |

*: Pap = Paper; Coc = coconut fibre; Poly = polystyrene; [1]: Total mesophilic aerobic bacteria; [2]: total aerobic psychrophilic bacteria; [3]: total *Enterobacteria*; [4]: *E. coli*; [5]: *Pseudomonas* spp.

Mesophilic and psychrophilic bacteria are important indicators of the hygienic quality and shelf-life of fish [31]. Contamination by mesophilic bacteria occurs during processing and packaging and can cause alterations in fish products. However, the bacterial flora of fish is dominated by Gram-negative psychrophilic microorganisms of the genera *Acinetobacter*, *Flavobacterium*, *Moraxella*, *Shewanella*, and *Pseudomonas*. Members of the families *Vibrionaceae* and *Aeromonadaceae* are common aquatic bacteria typical of the fish flora [32]. In addition, Gram-positive organisms, such as *Bacillus*, *Micrococcus*, *Clostridium*, and *Lactobacillus*, may be present in the fish flora in varying proportions. These microorganisms can easily alter fish products even at near-zero refrigeration temperatures [33].

The count of total *Enterobacteria* and *Pseudomonas* spp. Increased at the end of the storage period and was significantly greater ($p < 0.05$) in Pol stored samples as compared to Pap and Coc ones. The *E. coli* count was negative for all samples, regardless of the packaging material and storage time.

*Pseudomonas* spp. and *Enterobacteria* are widely recognized as specific spoilage organisms (SSO) of seafood products when they exceed 5 log cfu $g^{-1}$ and 3 log cfu $g^{-1}$, respectively [31,34,35]. In this study, *Pseudomonas* spp. and *Enterobacteria* loads were low in all the groups analyzed.

The hygienic quality of *Engraulis encrasicolus* packaged in both the organic Pap and Coc boxes proved to be better than the traditional Pol boxes. The poor hygienic quality of the *Engraulis encrasicolus* packed in Pol is probably due to the higher porosity of the packaging, which allows a higher bacterial proliferation. Similarly, in a study by Alzate et al. [36] on the use of an edible film based on hydroxypropyl methyl cellulose and glycerol, the surface hydrophobicity of the film improved antimicrobial activity against *Pseudomonas fluorescens* and other bacteria, preserving the shelf-life of the food.

In our study, the physical evaluation of the two organic boxes containing paper and coconut fiber showed good resistance up to seven days of storage. The Coc boxes were more resistant, whereas the Pap ones deformed, although they remained undamaged. Both the organic Pap and Coc boxes were lighter as compared to Pol ones and at the same time extremely robust, water resistant, and suitable for use in cold conditions.

### 3.2. Fatty Acid Profile of Anchovy Fillets

Table 2 reports the fatty acid profile of anchovy fillets during storage in different types of packaging. The most abundant saturated fatty acid is the palmitic acid (C16:0), a primary SFA in this species, which ranged from 26.10% to 30.00%, in accordance with previous findings [37,38]. In all the packaging groups, the C16:0 and C23:0 fatty acids showed a significant increase ($p < 0.05$) from T0 to T2. Therefore, the total saturated fatty acid ($\sum$ SFA) concentration also increased ($p < 0.05$) in Pol and Pap packaging. The fatty acid C23:0 showed a two-fold increase from T0 to T2 in all the packaging groups. A similar increase in the concentration of SFAs, found also by other authors, may be due to the occurrence of the oxidation process during storage, leading to the increase in the total amount of SFA and to the decrease in the total amount of UFA, thus affecting fish quality spoilage [39].

**Table 2.** Fatty acid profile (% Total Fatty Acid Methyl Esthers) in anchovies stored in relation to different packaging materials and storage times.

| | Pol | | | Pap | | | Coc | | | | Effects | | |
|---|---|---|---|---|---|---|---|---|---|---|---|---|---|
| | T0 | T1 | T2 | T0 | T1 | T2 | T0 | T1 | T2 | SEM | M | S | M × S |
| Total Fatty acids (g/100 g muscle) | 17.42 | 7.52 | 4.65 | 18.75 | 6.69 | 4.38 | 18.03 | 5.72 | 4.00 | 0.192 | 0.802 | 0.413 | 0.199 |
| $C_{14:0}$ (myristic) | 2.63 | 4.13 | 4.10 | 2.94 | 4.04 | 4.34 | 3.24 | 3.64 | 3.95 | 0.685 | 0.054 | 0.054 | 0.067 |
| $C_{15:0}$ (pentadecilic) | 0.78 | 0.96 | 0.95 | 0.84 | 0.91 | 0.98 | 0.84 | 0.90 | 0.80 | 0.140 | 0.172 | 0.155 | 0.246 |
| $C_{16:0}$ (palmitic) | 28.90 [b] | 29.10 [ab] | 31.10 [a] | 30.00 [b] | 30.74 [ab] | 31.47 [a] | 28.30 [b] | 28.50 [ab] | 30.10 [a] | 1.558 | 0.241 | 0.045 | 0.069 |
| $C_{17:0}$ (margaric) | 1.07 [a] | 0.99 [ab] | 0.70 [b] | 1.12 [a] | 0.90 [ab] | 0.81 [b] | 1.13 | 0.90 | 0.90 | 0.140 | 0.421 | 0.050 | 0.077 |
| $C_{18:0}$ (stearic) | 6.90 | 6.90 | 8.10 | 6.68 | 6.80 | 8.30 | 6.90 | 7.10 | 7.90 | 1.267 | 1.015 | 2.087 | 1.769 |
| $C_{21:0}$ (heneicosylic) | 0.20 | 0.20 | 0.21 | 0.43 | 0.30 | 0.30 | 0.50 | 0.50 | 0.82 | 0.173 | 0.874 | 0.895 | 0.799 |
| $C_{23:0}$ | 3.90 [b] | 8.90 [a] | 11.20 [a] | 4.10 [b] | 7.90 [a] | 8.70 [a] | 4.31 [b] | 8.10 [a] | 8.70 [a] | 1.853 | 0.147 | 0.039 | 0.065 |
| ∑ SFA [1] | 44.38 [b] | 51.18 [ab] | 56.36 [a] | 46.11 [b] | 51.59 [ab] | 54.90 [a] | 45.22 | 49.64 | 53.17 | 1.067 | 0.744 | 0.047 | 0.074 |
| $C_{16:1}$ (palmitoleic) | 1.68 | 1.48 | 1.10 | 1.68 | 1.02 | 1.21 | 1.58 | 1.02 | 1.08 | 0.439 | 0.210 | 0.445 | 0.394 |
| $C_{18:1n9}$ (oleic) | 12.07 [a] | 11.00 | 9.00 [b] | 12.08 [a] | 11.07 | 10.09 [b] | 13.09 [a] | 11.16 | 10.65 [b] | 1.490 | 0.753 | 0.044 | 0.145 |
| $C_{22:1n9}$ (cetoleic) | 1.24 [a] | 0.14 [b] | 0.07 [b] | 0.60 | 0.13 | 0.11 | 1.19 [a] | 0.12 [b] | 0.11 [b] | 0.483 | 0.199 | 0.031 | 0.049 |
| $C_{24:1n9}$ | 0.40 | 0.40 | 0.36 | 0.45 | 0.40 | 0.40 | 0.48 | 0.47 | 0.42 | 0.149 | 0.554 | 0.183 | 0.229 |
| ∑ MUFA [2] | 15.39 [A] | 13.02 [a] | 10.53 [Bb] | 15.53 | 12.62 | 11.81 | 16.34 [Aa] | 12.77 [b] | 12.26 [B] | 2.200 | 0.217 | 0.001 | 0.024 |
| $C_{18:2n6}$ (linoleic) | 2.50 | 2.47 | 2.17 | 2.01 | 2.00 | 1.94 | 2.40 | 2.35 | 1.78 | 0.280 | 0.147 | 0.065 | 0.071 |
| $C_{18:3n6}$ (γ-linolenic) | 0.95 | 0.95 | 0.90 | 1.01 | 1.06 | 1.00 | 0.80 | 0.80 | 0.77 | 0.099 | 0.361 | 0.185 | 0.146 |
| $C_{18:3n3}$ (α-linolenic) | 0.93 | 0.84 | 0.79 | 1.10 | 1.00 | 1.00 | 0.80 | 0.81 | 0.80 | 0.104 | 0.449 | 0.874 | 0.796 |
| $C_{20:3n6}$ | 0.84 | 0.80 | 0.63 | 0.98 | 0.80 | 0.77 | 0.95 | 0.89 | 0.84 | 0.118 | 0.741 | 0.895 | 0.844 |
| $C_{20:4n6}$ (ARA) | 0.80 | 0.81 | 0.61 | 0.89 | 0.85 | 0.61 | 0.80 | 0.82 | 0.70 | 0.106 | 0.845 | 0.726 | 0.476 |
| $C_{20:5n3}$ (EPA) | 10.98 [a] | 9.25 [b] | 8.13 [b] | 11.11 [a] | 9.56 [ab] | 8.23 [b] | 11.13 [a] | 10.74 [ab] | 10.15 [b] | 0.401 | 0.165 | 0.048 | 0.050 |
| $C_{22:5n3}$ (DPA) | 3.20 [a] | 2.70 [ab] | 2.10 [b] | 3.30 [a] | 2.70 [b] | 2.00 [b] | 3.05 [a] | 3.00 [ab] | 2.20 [b] | 0.472 | 0.264 | 0.031 | 0.045 |
| $C_{22:6n3}$ (DHA) | 18.40 [a] | 18.00 [ab] | 17.70 [b] | 18.04 [a] | 17.63 [ab] | 17.11 [b] | 18.62 [a] | 17.73 [b] | 17.11 [b] | 0.486 | 0.128 | 0.012 | 0.047 |
| ∑ PUFA [3] | 38.57 [a] | 36.92 [ab] | 35.49 [b] | 38.44 [a] | 35.60 [ab] | 32.66 [b] | 38.65 [a] | 37.12 [ab] | 34.35 [b] | 1.512 | 0.471 | 0.038 | 0.054 |
| n − 3 [4] | 33.51 [a] | 30.79 [ab] | 28.62 [b] | 33.55 [a] | 30.89 [ab] | 28.34 [b] | 33.50 [a] | 32.28 [ab] | 30.26 [b] | 1.290 | 0.763 | 0.047 | 0.078 |
| n − 6 [5] | 5.09 | 5.03 | 4.51 | 4.89 | 4.71 | 4.71 | 4.95 | 4.84 | 4.09 | 0.319 | 0.241 | 0.132 | 0.154 |
| n − 6/n − 3 | 0.15 | 0.16 | 0.16 | 0.15 | 0.15 | 0.15 | 0.15 | 0.15 | 0.15 | 0.007 | 0.133 | 0.249 | 0.188 |
| n − 3/n − 6 | 6.58 | 6.12 | 6.35 | 6.86 | 6.66 | 6.56 | 6.77 | 6.67 | 7.40 | 0.356 | 0.625 | 0.484 | 0.362 |
| AI [6] | 0.72 | 0.79 | 0.77 | 0.79 | 0.83 | 0.81 | 0.85 | 0.86 | 0.86 | 0.044 | 0.078 | 0.052 | 0.064 |
| TI [7] | 0.21 | 0.19 | 0.18 | 0.19 | 0.18 | 0.19 | 0.16 | 0.18 | 0.16 | 0.195 | 0.476 | 0.221 | 0.094 |

Pol: Polystyrene; Pap: Paper, Coc: Coconut fibre. Storage times: T0 = 1 day, T1 = 3 days and T2 = 7 days. SEM: Standard error of means; a, b: $p < 0.05$; A, B: $p < 0.01$; [1] ∑ SFA—saturated fatty acids (sum of C14:0 + C15:0 + C16:0 + C17:0 + C18:0 + C21:0); [2] ∑ MUFA—monounsaturated fatty acids (sum of C16:1 n9 + C18:1 n9+ C22:1 n9); [3] ∑ PUFA—polyunsaturated fatty acids (sum of n − 6 + n − 3); [4] Total n − 3 (sum of C18:3 n3+ C20:5 n3 + C22:5 n3 + C22:6 n3); [5] Total n − 6 (sum of C18:2 n6 + C18:3 n6 + C20:4 n6); [6] AI—atherogenic index; [7] TI—thrombogenic index; differences between the storage periods: A, B: $p < 0.01$; a, b: $p < 0.05$.

In response to an increase of SFAs during storage, a significant ($p < 0.05$) reduction of total MUFAs was recorded in the Pol and Coc packaging. This reverse increase and decrease of saturated and unsaturated fatty acids were found to represent useful indices of the freshness and decomposition of marinated fish during storage [39]. Among MUFAs, the oleic acid (C18:1n9) was the most abundant fatty acid present in anchovy fillets, whose concentration showed a significant ($p < 0.05$) decrease during storage in all the packaging groups. These results are similar to those reported by Turhan et al. [40], who analyzed anchovy fillets brined with vegetable antioxidant extracts. Moreover, the C22:1n9 fatty acid showed a significant decrease ($p < 0.05$) in Pol and Coc groups.

The concentration of total polyunsaturated fatty acids ($\sum$ PUFA) decreased significantly ($p < 0.05$) from T0 to T2 in all the groups, and a similar trend was reported also by Tufan et al. [13] in anchovies caught in the Turkish Black Sea during winter. In this study, the total PUFAs in fresh anchovies ranged from 38.44% to 38.65% in all the groups; this amount is approximately equal to 6 g/100 g of an edible portion of fresh anchovies, i.e., a greater content as compared to the recommended daily intake of 0.4 g of total PUFAs, accordingly to Testi et al. [41].

The average concentration of eicosapentaenoic acid (EPA) found at T0 in all groups was lower compared to the results reported by Tanakol et al. [37], probably due to the different sea area and to the seasonal changes of captured anchovies. In all the packaging groups, the EPA content significantly ($p < 0.05$) decreased from T0 to T2, in accordance with the findings of Turan et al. [38]. Docosapentaenoic acid (C22:5n3; DPA) is another important omega-3 fatty acid, and its concentration in the three packaging groups significantly ($p < 0.05$) decreased from T0 to T2, in agreement with previous research [37,39,42,43]. In this study, docosahexaenoic acid (C22:6n3; DHA) was the second most abundant PUFA in the fillets, as found for anchovies captured in Black and Mediterranean Seas [13], while other studies observed a variation of the content of docosahexaenoic acid in relation to the month of capture [44]. In human nutrition, DHA is an important fatty acid endowed with well documented preventive effects on health, being able to limit the occurrence of coronary and artery diseases [45]. The Italian C.R.E.A. (Council for Research in Agriculture and Analysis of the Agricultural Economy) guidelines suggest that the weekly requirement for an adult on a 2200 kcal diet is approximately 2500 mg of EPA and DHA [46]. This recommended value is widely fulfilled by the results of our study, since 100 g of an edible portion ensures an average intake of approximately 2800 mg of EPA and DHA.

No significant differences, neither among groups nor among storage times, were found for the $n-6/n-3$ ratio of anchovy fillets. The $n-6/n-3$ ratio is an important nutritional indicator of food healthiness. According to Simopoulos [47], the western diet is deficient in omega-3 fatty acids. Therefore, a greater consumption of food with a $n-6/n-3$ ratio at least of 1:1 is highly advisable. In this study, the $n-6/n-3$ ratio of anchovy fillets is very low, due to the high $n-3$ fatty acids concentration; this ratio falls within the range of 0.14–0.16, thus showing potential benefits for human health.

The indices of atherogenicity and thrombogenicity did not significantly differ among groups. These indices are strong markers for predicting the risk of atherosclerosis and platelet aggregation. Therefore, lower AI and TI rates implicate a greater protective potential for coronary artery disease. In the current study, AI rates ranged from 0.72 to 0.86 (Table 2), accordingly to other studies which reported similar results for AI in marine fish species [48,49]. The TI values (Table 2) fell within the range 01.6–0.21, which contains very low values as compared to previous findings [50], indicating that the consumption of these species is beneficial to human health [28].

### 3.3. Lipid Oxidation of Anchovies Fillets

Table 3 shows the MDA concentration in anchovy fillets during the storage period as assessed by the T-BARS test. Fillets from the Pol group showed a significantly ($p < 0.05$) higher MDA concentration as compared to the Pap samples in all the times of storage. Among the Pap group, a markedly ($p < 0.05$) greater MDA concentration was found at

the end of the storage period as compared to T0. Fillets stored in the Coc boxes showed intermediate values in terms of MDA concentration at all the times analyzed, although lacking statistical significance as compared to Pap and Pol groups. The oxidation of lipids in fatty fish is a major problem during storage and processing and has been well documented in many fish species [21]. A possible explanation behind this susceptibility to oxidation is attributable to the number of pro-oxidants, such as heme proteins (Hb), myoglobin, low molecular weight (LMW) transition metal complexes, and lipoxygenases [51]. In this study, anchovies stored up to seven days showed better freshness, in contrast with the findings reported by Simeonidou et al. [42], who observed an increase of the MDA concentration in seafood during storage. Increased vulnerability towards oxidation may be related to the very high unsaturated fatty acid content in this fish species. Oxidation activators start their action immediately after catching, causing oxidative rancidity as storage proceeds [19]. However, for all the packaging materials and storage times investigated in this study, the MDA values recorded were far below the concentration of 2 mg MDA/kg meat, which is the limit above which rancidity is perceived by consumers [52].

**Table 3.** Results of TBARs values (mg malonaldehyde/kg fish muscle) in fillets in relation to different packaging materials and storage times.

| | \multicolumn{9}{c}{Packaging Materials} | SEM [1] | \multicolumn{3}{c}{Effects} |
| | \multicolumn{3}{c}{Pol} | \multicolumn{3}{c}{Pap} | \multicolumn{3}{c}{Coc} | | | | |
| | T0 | T1 | T2 | T0 | T1 | T2 | T0 | T1 | T2 | | M | S | M × S |
|---|---|---|---|---|---|---|---|---|---|---|---|---|---|
| MDA (mg/kg fillet) | 0.027 [x] | 0.032 [x] | 0.033 [x] | 0.016 [a,y] | 0.021 [ab,y] | 0.026 [b,y] | 0.018 [xy] | 0.024 [xy] | 0.027 [xy] | 0.006 | 0.038 | 0.044 | 0.097 |

Pol: Polystyrene; Pap: Paper, Coc: Coconut fibre. Storage times: T0 = 1 day, T1 = 3 days and T2 = 7 days. [1]: Standard error of means. Differences between materials: x; y: $p < 0.05$; differences between the storage periods: a, b: $p < 0.05$.

The higher concentration of MDA at T2 can be related to a higher oxidation of the MUFAs and PUFAs and to the enzymatic hydrolysis of anchovy lipids during storage. Similar decreases in MUFA and PUFA concentrations were observed by Simeonidou et al. [42] for sardines (*Sardine mediterraneus*), Atlantic mackerel (*Scomber scombrus*), and other Mediterranean filleted fish species during frozen storage. Moreover, the gradual increase of *Pseudomonas* spp. and the mesophilic and psychrophilic bacteria during the storage period, observed for all the packaging materials, may also be associated with a gradual loss of freshness, as demonstrated by the concomitant increase in the T-BARS value, especially at T2 [31,53–55].

## 4. Conclusions

This study provides preliminary information on the potential use of organic packaging materials for the storage of fishery products, by studying the degree of lipid oxidation and the change of fatty acid profile of anchovy fillets at different times of storage. The T-BARS test showed that fish storage in paper packaging determined a lower MDA concentration at all storage times in comparison with the traditionally used polystyrene, while coconut fiber packaging displayed intermediate results in terms of MDA concentration. In our study, all the materials used for packaging guaranteed a high quality of fish, since the amount of total n − 3 fatty acids and the atherogenicity and thrombogenicity indices were unaffected by the packaging system, providing fish fillets characterized by an optimal n − 6/n − 3 ratio, with benefits for human health. The microbiological results showed an improved hygienic quality of *Engraulis encrasicolus* following the use of the two organic paper and coconut fiber boxes due to the lower porosity of these packaging materials, which leads to lower microbial proliferation, and, as a consequence, to improved fish preservation. Even though the fish quality traits underwent microbiological and nutritional changes during storage from the first to the seventh day, it may be stated that anchovy quality is adequately preserved during refrigeration up to seven days.

In conclusion, the use of organic materials, such as coconut fiber and paper, for fish handling and storage is worthy of further investigation since these packaging materials did not affect the shelf-life and fatty acid profile of anchovies during storage. The use of alternative materials to plastic is strongly advisable given the potential contribution to the reduction of environmental pollution and impacts on climate change, and to the protection of marine ecosystems.

**Author Contributions:** Conceptualization, A.D.; Methodology, A.D., N.C.Q. and F.G.; Software, F.C.; Formal analysis, S.T.; Investigation, N.C.Q. and S.T.; Resources, F.G. and A.C.J.; Data curation, M.A.C. and S.T.; Writing—original draft, M.A.C. and S.T.; Supervision, M.R. Project administration, M.R.; Funding acquisition, A.C.J. All authors have read and agreed to the published version of the manuscript.

**Funding:** This work was supported by Regione Puglia (ITALY)—Progetto FEAMP "Salvaguardia di piccoli pelagici: Una pesca sostenibile ed innovativa nel Basso Adriatico" (SALV.ADRI); CUP: B91B17001140009.

**Institutional Review Board Statement:** Our research does not fall under the legislation for the protection of animals used for scientific purposes, national decree-law 113/2013 (2010-63-EU directive).

**Informed Consent Statement:** Not applicable.

**Data Availability Statement:** The data that support the findings of this study are available from the corresponding author upon reasonable request.

**Acknowledgments:** The authors express their gratitude to the technicians of the Department of Soil, Plant and Food Sciences Massimo Lacitignola, Nicolò Devito and Domenico Gerardi for their laboratory assistance.

**Conflicts of Interest:** The authors declare no conflict of interest.

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
