# Peer review of "Shelf-Life and Quality of Anchovies (Engraulis encrasicolus) Refrigerated Using Different Packaging Materials"

_fishes, doi:10.3390/fishes8050268_

Round 1

Reviewer 1 Report

The current form of the submitted manuscript (fishes-2384439) is not acceptable for publication in the journal of Fishes unless it is majorly revised. My comments and suggestion for revision are as follows.

Abstract

Line 11: to storage fish à to store fish, storage of fish?? Check for the correct grammar

# Mention the results of fatty acids in the Abstract

# Briefly explain the main conclusion of the study, limitation of the study, future study, potential application, etc, at the end of Abstract.

Introduction

Lines 73-74: Move the full name for EPA and DHA to line 64.

Materials and Methods

Lines 97-100: Provide the specification of packaging materials used in this study, for instance the size (length, width, height, thickness, etc) of the materials.

Lines 101-102: Did the authors measure the real temperature of the sample before and after re-icing? Uncontrolled re-icing might cause temperature varied between sample during storage, thereby leading to false results.

Results and Discussion

Lines 176-179: provide discussion for CAPT results from coconut group.

Line 179: log ufc à log cfu

# Table 1 and Figure 1 are derived from the same data. Therefore, choose one to prevent repetition.

Lines 200-202: discuss the results for Enterobacteria with reference

Lines 225-227: oxidation process of what?? Do you mean oxidation of MUFA and PUFA led to SFA composition in total fatty acids increase?

Lines 261-266: do you mean lower n-6/n-3 ratio means the samples are rich in omega 3, so it has potential health benefits? If so, the paragraph should be revised accordingly.

Lines 267-269: Discuss the results for AI and TI with reference whether the value is good or not according to health standard.

# Table 2: in this table, the authors only show the results with statistical significance, however the authors need to show all statistical significance (a,b; A,B) for all the data/results selected. For example, the statistical significance for C16:0 of Pol group samples were only shown for T0 and T2 while that for C23:0 were shown for all the storage periods. Therefore, check again carefully and provide all the statistical significance for all the selected data/results.

# Table 2: under the table, explain the comparison of the data is between materials or between storage periods? Just like the authors explain for Table 3

# Table 2: IA à AI

# Table 3: provide statistical significance for all the storage periods in Coc group samples.

Lines 287-303: Results for Coc samples should be discussed.

Lines 304-311: Provide the results of correlation analysis to show the relationship between TBARs (Table 3) and changes in MUFA/PUFA (Table 2) and microbial population (Table 1).

Conclusions

Results for microbial changes should be mentioned.

Author Response

Dear Reviewer,

Thank you for your valuable suggestions that have been very useful to improve the quality of our manuscript.

We have reported below the answers to your comments, hoping they are clear and exhaustive.

Best regards.

Abstract

Line 11: to storage fish à to store fish, storage of fish?? Check for the correct grammar

done

# Mention the results of fatty acids in the Abstract

done

# Briefly explain the main conclusion of the study, limitation of the study, future study, potential application, etc, at the end of Abstract.

done

 Introduction

Lines 73-74: Move the full name for EPA and DHA to line 64.

done

 Materials and Methods

Lines 97-100: Provide the specification of packaging materials used in this study, for instance the size (length, width, height, thickness, etc) of the materials.

Thank you for the suggestion, we added these information

Lines 101-102: Did the authors measure the real temperature of the sample before and after re-icing? Uncontrolled re-icing might cause temperature varied between sample during storage, thereby leading to false results.

Thank you for your comment. The temperature was constantly monitored by using a thermometer, both in the storage refrigerator and in the sample boxes. No temperature fluctuations greater than 0.5-1°C were recorded. In the Materials and methods section, we have specified more in detail.

 Results and Discussion

Lines 176-179: provide discussion for CAPT results from coconut group.

Thank you for the suggestion, we explained the result better for the Coc group.

Line 179: log ufc à log cfu

Thank you. We have corrected the word.

# Table 1 and Figure 1 are derived from the same data. Therefore, choose one to prevent repetition.

Thank you. We have deleted Figure 1

Lines 200-202: discuss the results for Enterobacteria with reference

Thank you. We have re-worded the sentence.

Lines 225-227: oxidation process of what?? Do you mean oxidation of MUFA and PUFA led to SFA composition in total fatty acids increase?

We added the explanation into the text.

Lines 261-266: do you mean lower n-6/n-3 ratio means the samples are rich in omega 3, so it has potential health benefits? If so, the paragraph should be revised accordingly.

We revised the paragraph according to the reviewer suggestion.

Lines 267-269: Discuss the results for AI and TI with reference whether the value is good or not according to health standard.

We revised the paragraph according to the reviewer suggestion.

# Table 2: in this table, the authors only show the results with statistical significance, however the authors need to show all statistical significance (a,b; A,B) for all the data/results selected. For example, the statistical significance for C16:0 of Pol group samples were only shown for T0 and T2 while that for C23:0 were shown for all the storage periods. Therefore, check again carefully and provide all the statistical significance for all the selected data/results.

We corrected the table 2 according to the reviewer suggestion.

# Table 2: under the table, explain the comparison of the data is between materials or between storage periods? Just like the authors explain for Table 3

We have checked the significance superscipts.

# Table 2: IA à AI

We have changed.

# Table 3: provide statistical significance for all the storage periods in Coc group samples.

We corrected the table 3 according to the reviewer suggestion.

Lines 287-303: Results for Coc samples should be discussed.

We added the comments into the text.

Lines 304-311: Provide the results of correlation analysis to show the relationship between TBARs (Table 3) and changes in MUFA/PUFA (Table 2) and microbial population (Table 1).

 Conclusions

Results for microbial changes should be mentioned.

Thank you for your suggestion, we added a sentence into the conclusion section.

Author Response

Dear Reviewer,

Thank you for your valuable suggestions that have been very useful to improve the quality of our manuscript.

We have reported below the answers to your comments, hoping they are clear and exhaustive.

Best regards.

  1. In the introduction, it might be more appropriate to start with the text that begins on line 51 and then introduce the texts that are in the first two paragraphs.

We have accepted your suggestion and changed the introduction.

  1. Line 95: which was the ratio fish/ice? And which was the type of ice (e.g., flake ice)

Thank you. We used flake ice, that was added until the sample was completely covered.

  1. Line 101: The Storage temperature selected was 0 °C (± 0.5 °C). Any reason for this temperature. Usually iced fish is kept at temperatures around 4 °C

We have kept fish in ice in accordance with the needs of microbiological analysis.

  1. In Figure 1, please see the names of packaging materials and indicate in the text the meaning of Pol: Polystyrene; Pap: Paper, Coc: Coconut fibre.

Usually, the legend of the figures follows the figure.

Thank you. Figure 1 was removed because it was repetitive as suggested by Reviewer 1.

.

  1. Line 216: It is said that “Both the organic Pap and Coc boxes were lighter as compared to Pol ones and at the same time extremely robust, water resistant and suitable for use in cold conditions”.

Which was weight of the different boxes?

How the water resistance was measured?

Thank you. The assessment on lightness, strength, and water resistance are based on subjective evaluations so we do not have data on measurements.

  1. The MDA contents are quite low and the selected Storage temperature 0 °C (± 0.5 °C) may have contributed for such values. The authors should discuss the results achieved with those of other authors obtained under more usual ice Storage conditions.

We have discussed our results as compared to those reported by Simeonidou et al. 1997 who managed fish kept at 1 °C, so in temperature conditions similar to ours.

Round 2

Reviewer 1 Report

In the first review, my comments and suggestion for discussion in Lines 320-327 were not addressed by the authors.

Before publication, discussion (Lines 320-327) needs to be improved.

Lines 322: Similar decreases of what? make it clear

Lines 324-327: provide findings from previous published papers that show the increase in the bacterial population (Pseudomonas spp., mesophilic and psychrophilic bacteria) during storage is associated with the gradual loss of freshness or increase in TBAR-S value.

Author Response

Dear Reviewer,

Thank you for your valuable suggestions. We have corrected the paraghaph and added the references as request.

best regards